

# An update on dissolved methane distribution in the North subtropical Atlantic Ocean

Anna Kolomijeca[1,2] Lukas Marx[3], Sarah Reynolds[3], Thierry Cariou[1,4], Edward Mawji[5], Cedric Boulart[1]

[1] UMR 7144 CNRS Sorbonne Université, Station Biologique de Roscoff, 29680 Roscoff, France

[2] MARUM, Center for Marine Environmental Sciences, D-28359 Bremen, Germany

[3] School of the Environment, Geography and Geosciences, University of Portsmouth, PO1 2UP Portsmouth, UK

[4] IRD, UAR191, Instrumentation, Moyens Analytiques, Observatoires en Géophysique et Océanographie (IMAGO), Technopôle de Brest-Iroise, 29280 Plouzané, France

[5] National Oceanography Centre, European Way, Southampton, SO14 3ZH, UK

*Correspondence to*: Anna Kolomijeca (akolomijeca@marum.de)

**Abstract.** Methane ($CH_4$) is the second most produced greenhouse gas after carbon dioxide, however the role of the open ocean in its natural cycle remains poorly constrained. Accumulating evidence indicates that a significant part of oceanic $CH_4$ is produced in oxygenated surface waters as a by-product of phytoplanktonic activity. The subtropical North Atlantic Ocean

between 26ºN 80'W and 26ºN 18'W was investigated for the distribution of dissolved $CH_4$ concentrations and associated air-sea fluxes during winter 2020. Water samples from 64 stations were collected from the upper water column up to depths of 400 m. The upper oxic mixed-layer was oversaturated in dissolved $CH_4$ with concentrations ranging between 3-7 nmol/l, with the highest values of 7-10 nmol/l found to the east of the transect, consistent with other subtropical regions of the world's oceans. The high anomalies of dissolved $CH_4$ appeared to be associated to phosphorus depleted waters and to a peak of regions

of elevated phytoplankton abundance. Further investigations indicated a correlation between $CH_4$ anomalies, phosphate depletion and the abundance of two ubiquitous pico-cyanobacteria, *Synechococcus* and *Prochlorococcus*, although other phytoplanktonic phyla cannot be excluded. The calculation of air-sea fluxes confirms the subtropical North Atlantic Ocean as a source of $CH_4$, mainly produced by phytoplanktonic activity in surface waters.

## 1. Introduction

Since the industrial revolution, the average global temperature has increased at the fastest rate in recorded history, primarily driven by growing emissions of greenhouse gases (GHGs). Among them, methane ($CH_4$) is considered the second largest contributor to Earth warming, after carbon dioxide ($CO_2$), with an atmospheric concentration of 1,866 ppb (IPCC, 2021). Over the last 50 years, $CH_4$ concentrations have increased by 20 % (Karl et al., 2008), (Rhee et al., 2009) – and is expected to rise further, by approximately 2 % per year (Dang and Li, 2018).

Oceans are generally thought of as a minor contributor to the total global $CH_4$ budget, however, recent calculations indicate that oceans could emit 6 to 17 Tg CH4/yr, i.e. 1 to 10 % of the total natural emissions (Weber et al., 2019). This large variability



reflects the great uncertainty on the contribution of natural sources due to a lack of data, when compared to carbon dioxide ($CO_2$). $CH_4$ is traditionally thought to be produced by microbial anaerobic methanogenesis in marine sediments as a consequence of the degradation of organic matter; it accumulates in the sediment, eventually forming gas hydrates, which may

then be released into the water column. Under the influence of pressure and temperature $CH_4$ diffuses out of the sediment and ebullition carries $CH_4$ to the atmosphere (Weber et al., 2019).

The marine flux of $CH_4$ results from the balance between production and oxidation processes, as for instance the microbial anaerobic oxidation of $CH_4$ (AOM) in sediments significantly decreases $CH_4$ fluxes to the atmosphere, thus representing an important carbon sink in the ocean (Oppo et al., 2020). In fact, the marine flux is dominated by shallow coastal environments

including estuaries (up to 75 % (Weber et al., 2019). In the open oxygenated waters, the primary mechanism controlling the $CH_4$ emissions is aerobic methanotrophy that converts $CH_4$ into $CO_2$ (Weber et al., 2019). However, this process may be overcome by *in-situ* production of $CH_4$ in upper oxic waters that can significantly contribute to marine $CH_4$ fluxes to the atmosphere. Typical $CH_4$ depth distribution in the open ocean indicates a general oversaturation in the mixed layer (Reeburgh, 2007a); In the surface waters of the Pacific Ocean (Weller et al., 2013), the Indian Ocean (Bui et al., 2018) and the Atlantic

Ocean, values of 2-5 nmol/l and a maximum of 10 nmol/l were measured near the surface (Scranton and Brewer, 1977). These observations make the global ocean a net source of $CH_4$ for the atmosphere with a weighted supersaturation of 120 % (Kock and Bange, 2007). Exemptions are the Southern Ocean and the central Arctic Ocean, where surface waters are undersaturated in $CH_4$, either due to extensive upwelling supplying $CH_4$-depleted water to the surface (Bui et al., 2018), or the limitation of air-sea exchanges by ice cover (Weber et al., 2019).

The oversaturation of the surface mixed layer, commonly known as the "ocean methane paradox", was initially described as a result of the methanogenic activity from *Archaea* living within anaerobic cavities of the zooplankton gut and anaerobic environments inside sinking particles (Reeburgh, 2007a). Initially, only microbes from the *Archaea* domain were thought to have the capabilities of producing $CH_4$ under strict anaerobic conditions. Although one cannot exclude this process to explain the methane paradox (Schmale et al., 2018) (Stawiarski et al., 2019), an increasing number of studies focus on the relationship

between $CH_4$ anomalies in surface waters and the presence of phytoplanktonic groups such as *coccolithophores* (Lenhart et al., 2016a) or cyanobacteria (Bižić et al., 2020).

Cyanobacteria are ubiquitous to every aqueous environment on Earth, both in illuminated and dark water bodies (Percival and Williams, 2013). In the open ocean, small-sized picophytoplankton of the genera *Prochlorococcus spp.* and *Synechococcus spp.* account for ~80 % of the total phytoplanktonic chlorophyll *a* (Hickman et al., 2010) and could represent up to 8.5 and

16.7 % of the ocean net primary production (ONPP), respectively (Flombaum et al., 2013). Generally, nutrient limitation sets the upper limit for primary production and the distribution of *Prochlorococcus* and *Synechococcus*; The oligotrophic subtropical North Atlantic in fact is nitrogen (N) - phosphorus (P) co-limited (Harvey et al., 2013), hence cyanobacteria need to acquire these nutrients from alternative sources. Whereas biological nitrogen fixation by diazotrophs supplies globally 163.2 Tg new nitrogen per year (Wang et al., 2019), *Prochlorococcus* and *Synechococcus* mostly depend on the remineralisation of

dissolved organic phosphorus (DOP) via hydrolytic enzymes (e.g. alkaline phosphatase (Muñoz-Marín et al., 2020).



Additionally, evidence is mounting that cyanobacteria are major sources of semi-labile dissolved organic matter (DOM) phosphonates (Repeta et al., 2016). The bacterial degradation of methylphosphonates (MPn) releases $CH_4$ and therefore cyanobacteria might play a key role in the global marine $CH_4$ flux. Laboratory-based studies with *Prochlorococcus* and *Synechococcus* confirmed a substantial production of $CH_4$ of up to $0.51 \pm 0.26$ µmol $g^{-1}$ $hour^{-1}$ (Bižić et al., 2020).

However, *in-situ* production of $CH_4$ is difficult to assess, due to the complexity of biogeochemical and physical processes involved in the marine $CH_4$ flux. Additionally, nutrient availability impacts the metabolic pathways of the cyanobacterial community.

$CH_4$ *in situ* production results from different metabolic pathways including the conversion of methylated substrates, which are
induced by environmental stress (e.g. nutrient supplies, temperature variations, light). As such, $CH_4$ may be the byproduct of the methylphosphonate decomposition in phosphate-stressed surface waters, i.e. the MPn way (Karl et al., 2008, Bizic et al 2020). Due to the strong depletion of inorganic phosphorus in some oligotrophic areas in the Atlantic or the Pacific Oceans, cyanobacteria use the organic phosphonates as a P source, leading to the release of methyl groups in the water that are rapidly converted into $CH_4$.

In contrast, nitrate availability might control CH4 production in phosphate replete surface waters. While in P-limited waters cyanobacteria use methylphosphonate as a nutrient source and hence release CH4, in N-limited waters, CH4 may result from the breakdown of DMSP (dymethylsuffoniopropionate) into DMS by bacteria as they use DMSP as a C source (Florez-Leiva et al., 2013). Other environmental parameters, such as variations in temperature or light attenuation may also influence $CH_4$ formation, although data is lacking to fully understand the metabolic pathways leading to $CH_4$ production.

In the current context of climate change and resulting consequences to the marine environment, we are likely to expect a shift in the primary producers towards smaller sized cyanobacteria such as *Prochlorococcus* and *Synechococcus* (van de Waal and Litchman, 2020), with a concurrent decrease in total biomass in the open ocean (Marinov et al., 2010). Yet, with accumulating evidence of the importance of the cyanobacterial contribution towards $CH_4$ production and given the important role of $CH_4$ as a potent GHG, it is crucial to intensify the monitoring and investigating of $CH_4$ production and fluxes in surface oceanic waters
in order to feed a global database (Bange et al., 2009). There is a lack of data to comprehend the current and future role of the oceans in the global $CH_4$ budget.

In the present paper, we provide an update on the dissolved $CH_4$ concentrations and air-sea fluxes of $CH_4$ in surface waters of the subtropical North Atlantic Ocean as part of the JC191 hydrographical cruise (RRS James Cook, Jan-Mar 2020). Furthermore, we present data of the distribution of the two dominant cyanobacteria, *Prochlorococcus* and *Synechococcus* to
highlight the contribution of the planktonic community to the $CH_4$ flux. More specifically, our objectives were to confirm the presence of $CH_4$ anomalies in surface waters and the associated air-sea fluxes, and to propose possible mechanisms and sources controlling the $CH_4$ distribution by examining the relationships between the physico-chemical and the biological parameters.



## 2. Methods

### 2.1 Water sampling and analysis for dissolved gas concentrations

Seawater samples were collected during the JC191 hydrographical cruise (as part of the GO-SHIP program, PI A. Sanchez-Franks (Sanchez-franks, 2020)) on board the RRS James Cook between January-March 2020 along a west-east transect in the subtropical North Atlantic from Fort Lauderdale, USA to Santa Cruz de Tenerife, Spain on the nominal 24°N parallel. 64 profiles (out of 135 stations occupied by CTD casts in total) from surface to 400 m depth (or full-depth for the shallower

continental margin stations) were sampled (Fig. 1).

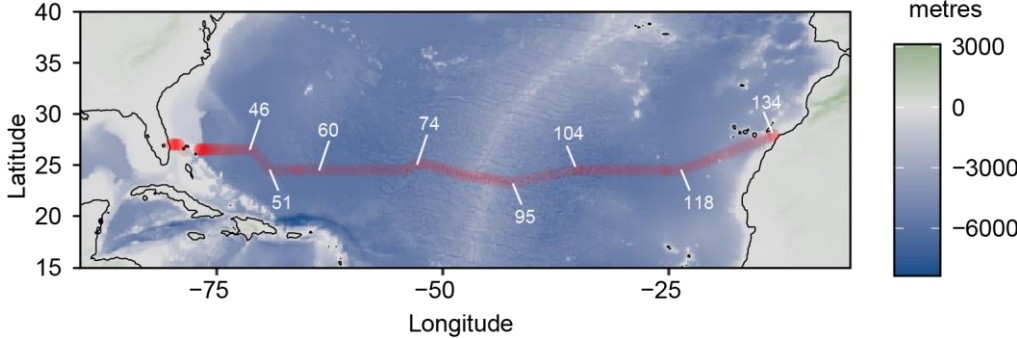

**Figure 1. Cruise track of JC191 24N expedition. White numbers indicates some of the CTD stations.**

Water samples for dissolved CH$_4$ measurements were collected into 20-ml headspace vials using a 24-Niskin Bottle rosette

equipped with a SBE911+ CTD (Conductivity, Temperature, Depth) and a Dissolved Oxygen Sensor (Seabird SBE 43). Dissolved gas samples were poisoned with Hg(Cl)$_2$, then fitted with Teflon stopcocks and crimp-sealed under exclusion of any air bubbles. Immediately after sampling, samples were stored at 4 °C until analysis on shore at the Station Biologique de Roscoff.

Gas extraction and analyses were performed using a Shimadzu Headspace Sampler (HS-20) connected to a gas chromatograph

(Shimadzu GC-2030), fitted with a barrier discharge ionization detector (BID) and a 30-m SH-Rt-MSieve 5A column. With this set-up, headspace extraction is entirely automated with pressurization of the sample up to 2 bars, heating at 90 °C and equilibration for 7 minutes. An aliquot of the gas sample was transferred to a 1-ml injection loop, maintained at 150 °C and injected into a 50 °C heated column. Calibrations were made by injecting known volume of standard gas (Messer®, 1000 ppm, 500 ppm and 100 ppm (-+ 1ppm), CH$_4$ in helium and 500 ppb H$_2$ in helium). All analyses were made in duplicate and results

are given as averaged values. The detection threshold of this method is 0.2 nmol for dissolved CH$_4$ and variation between duplicates was 5%.



## 2.2 Inorganic Nutrient Analysis

Samples for inorganic nutrient analysis ($NO_2^-$, $NO_3^-$, $NH_4^+$, $PO_4^{3-}$) were collected unfiltered into sterile 15 ml centrifuge
tubes (rinsed three times with water from the same Niskin). Samples were analysed directly from the collection tubes within
1-8 hours and measured from the lowest to the highest concentration (surface to deep) to reduce any carry over effects.
Nutrients were analysed on board using a 4-channel Seal Analytical AA3 segmented flow autoanalyzer following GO-SHIP
protocols (Becker et al., 2020))

In order to test the accuracy and precision of the analyses, certified reference materials (CRMs) from KANSO Technos Co.
(lots lot CD, CJ, CI and lot BW) were measured in triplicate in every run.

## 2.3 Cyanobacteria sampling and analysis

38 of the stations occupied during JC 191 were sampled at six depths in the upper water column (max. sampling depth 375 m)
following the live fluorescence profiles (AquaTracka III, Chelsea Technologies), to determine the prevailing community of
primary producers (unpublished data, Marx 2020). Bulk water samples (5 L) were collected from Niskin bottles, from which
then subsamples for the flow cytometric determination of *Prochlorococcus* and *Synechococcus* abundances were collected. 4
ml of sample water were immediately fixed with 40 µl Glutaraldehyde solution (50%) and stored at 4°C until transferred to a
low temperature freezer (-80 °C) after 12 hours.

Samples were analysed at the University of Portsmouth on a CyFlow Cube 8 (Sysmex) flow cytometer immediately after
defrosting and at a flowrate of 1 µl/s. The distinction between *Prochlorococcus* and *Synechococcus* was achieved by gating
each group according to its fluorescence signals (red and orange fluorescence) against the size fractionation (forward and side
scatter). For each of the stations sampled, the mixed layer depth (MLD, defined as the depth at which temperature decreased
by 1 °C from the surface) was determined and the integrated average in abundance above said MLD was calculated.

## 2.4 Statistical analyses

In order to determine the biological and physico-chemical parameters that influence the distribution of dissolved methane in
surface waters, a principal component analysis (PCA) was applied. This statistical tool simplifies the underlying structure of
the multivariate dataset converting a large number of variables into a smaller number of variables, i.e. components (PCs) with
a minimum loss of information. Each PC is usually associated to an eigenvalue that indicates the variation in the data. Here,
the PCA was performed using RStudio and the R 'stats' package' (Foundation for Statistical Computing, Vienna, 2013), in
which the calculation is done by a singular value decomposition of the data matrix, not by using eigen on the covariance matrix.
This method is preferred for numerical accuracy. Factor loadings were then calculated: high factor loadings indicate significant
correlation between variables.





In addition to the PCA, we ran simple correlation tests to evaluate the association between two variables (e.g. methane concentrations and fluorescence or phytoplankton abundances) for some stations. We used here the Kendall rank correlation
coefficient as the data do not follow a bivariate normal distribution according to the Shapiro-Wilk test.

**2.5 Flux calculation**

The flux of air-sea $CH_4$ was calculated following established methods based on (Kelley and Jeffrey, 2002a; Wanninkhof, 2014)

$$F = k (C_w - C_a)$$


Where, F is $CH_4$ flux (mol m$^{-2}$ d$^{-1}$) from sea water to air; k – gas transfer coefficient, related to wind speed; $C_w$ – average measured $CH_4$ concentration in the surface water, and $C_a$ – air-balancing (equilibrated) sea water $CH_4$ concentration. The wind speed data were obtained during the cruise and $CH_4$ concentration in the air was assumed to be 1.9 ppm (based on NOAA, Global Monitoring Laboratory https://www.esrl.noaa.gov report).
Gas transfer coefficient k, was calculated based on the relationship provided by (Wanninkhof, 2014):

$$k = 0.251 <U^2> (Sc/660)^{-.05}$$

Where, the value of 0.25 1 is obtained from inverse modelling approach, the CCMP winds, and the Modular Ocean Model-
General Circulation model, MOM3 GCM, as proposed by (Wanninkhof, 2014); <U2> is average of neutral stability winds at 10 m height squared, or the second moment. Sc is Schmidt number, which is kinematic viscosity of the water divided by the molecular diffusion coefficient of the gas.

Sc was calculated as $Sc = A + Bt + Ct^2 + dt^3 + Et^4$

Coefficients (A, B, C, D and E) for a least squares fourth-order polynomial fit were obtained from (Wanninkhof, 2014).

## 3. Results

### 3.1 Spatial distribution of physico-chemical parameters

$CH_4$ concentrations in surface waters of the North Atlantic ocean along the 24°N parallel were distributed non-uniformly between 3-10 nmol/l, i.e. systematically above saturation of ~2.7 nmol/l. Lowest concentrations of $CH_4$ of 3-4 nmol/l were found in the central gyre system above the mid-Atlantic ridge (~ 45 °W), with increasing values of 8-10 nmol/l in both western and eastern boundaries closer to the continental shelf (Figure 2a).



Chlorophyll *a* (Figure 2b, Chl *a*) from real-time fluorescence profiles, exhibit highest concentrations near both western and
eastern shore systems, with lowest concentrations in surface waters throughout the central gyre system (< 0.1 µg/l, Figure 2).
Nonetheless, deep chlorophyll maxima (DCM) were observed sitting right above MLD at ~ 100-130 m water depth.
Concentrations of Chl *a* increase towards the Mauritanian upwelling off the North African coast to above 0.4 µg/l, indicating
higher primary production due to enhanced nutrient supply. Accordingly, light transmission is decreased due to higher content
of suspended particles in the water column (Figure 2h). Furthermore, fluorescence does align with patterns of dissolved oxygen
in the water column of this transect. The surface waters in the subtropical North Atlantic are well oxygenated with
concentrations above 200 µmol/kg in the top 100 metres of the water column, subsequently decreasing with depth. Also,
increased concentrations of dissolved oxygen are observed towards the eastern boundary following enhanced lateral transport
from the coastal upwelling (Figure 2e).

Phosphate and nitrate concentrations were low (< 0.1 µmol/kg, <0.01 µmol/kg, respectively) in surface waters throughout the
transect with nutriclines >200 m depth, shallowing (<150 m in eastern basin and <100 m in eastern boundary) towards the
eastern boundary due to coastal upwelling off the eastern North African coast and associated enhanced mixing of deep, nutrient
enriched waters into surface waters (Figure 2c and d).




**Figure 2. Distribution of a) CH₄, b) Chl *a*, c) NO₃, d) PO₄, e) dissolved oxygen, f) salinity, g) temperature and h) transmission along
the cruise track. Chl *a*, dissolved oxygen, salinity, temperature and transmission data originate from sensors of and attached to the
CTD. CH₄, NO₃ and PO₄ were measured analytically. Part of the data were taken from CCHDO (Sanchez-franks, 2020).**

### 3.2 Sea-to-air methane flux

At most stations, air-sea $CH_4$ flux was 1-2 µmol $m^{-2}$ $d^{-1}$, with maximum value of 5 µmol $m^{-2}$ $d^{-1}$ in the area of 24 °N 25 °W
(Figure 3). Only one station (~24 °N 40 °W) had a negative $CH_4$ flux, which indicates a sink of $CH_4$. The overall average flux
is 1.9 µmol $m^{-2}$ $d^{-1}$. Note, that air-sea flux of $CH_4$ depends on the wind speed, which means that values can change depending
on location specific weather conditions.



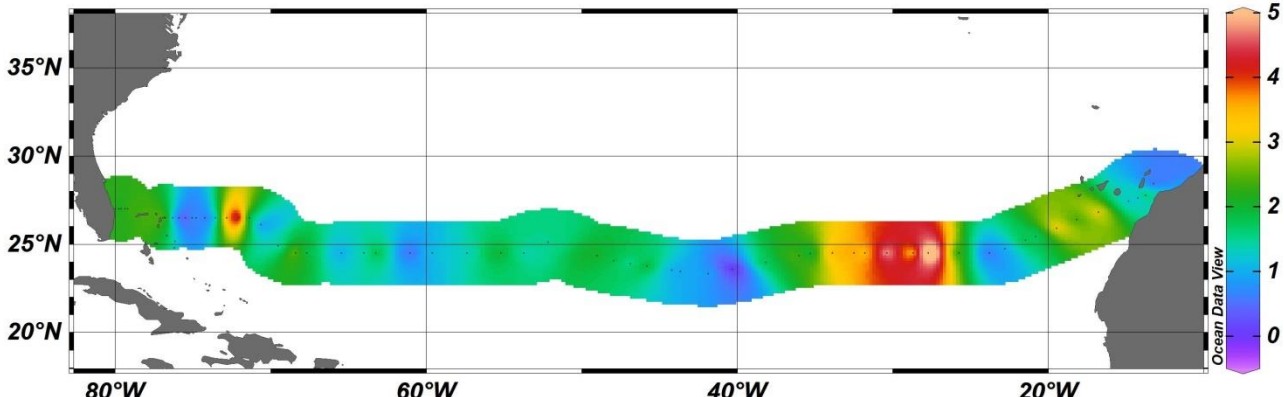

**Figure 3. Sea-to-Air CH$_4$ flux, µmol m$^{-2}$ d$^{-1}$.**

Other studies report similar values for of 1.6 µmol m$^{-2}$ d$^{-1}$ sea-to-air CH$_4$ flux in the North Pacific. However this can differ
significantly in other places, for example, sea-to-air CH$_4$ flux is 1.6-4.4 µmol m$^{-2}$ d$^{-1}$ in Sargasso Sea (Holmes et al., 2014); 1-
160 µmol m$^{-2}$ d$^{-1}$ in Belgic coastal zones (Borges et al., 2016); 13.3 µmol m$^{-2}$ d$^{-1}$ in the Red Sea mangroves (Sea et al., 2018);
6.5-7.4 4 µmol m$^{-2}$ d$^{-1}$ in east China Sea (Ye et al., 2015); and 0.38 µmol m$^{-2}$ d$^{-1}$ in Golf of Mexico (Kelley and Jeffrey, 2002b).

## 4. Discussion

215    **4.1 Methane distribution in surface waters of the subtropical North Atlantic Ocean**

Our data across the subtropical North Atlantic Ocean unambiguously indicates an oversaturation in CH$_4$ of the surface layer
(400 m) of 110 to 370%, which is in agreement with previous observations describing concentrations varying between 2-5
nmol/l with maximum of 10 nmol/l (Scranton and Brewer, 1977; Conrad and Seiler, 1988; Forster et al., 2009; de la Paz et al.,
2015; Leonte et al. , 2020). This is also in line with previous observations that describe the upper oceanic layer as a source of
220    CH$_4$ to the atmosphere (Reeburgh, 2007b; Dang and Li, 2018), in subtropical areas of the global ocean and also in some regions
of the Arctic Ocean (Kitidis et al., 2010; Kudo et al., 2018). To date, only the Southern Ocean is undersaturated in CH$_4$,
although this can be due to the scarcity of data collected, which needs to be complemented (Bui et al., 2018).

The distribution of dissolved CH$_4$ was variable across the North Atlantic with higher concentrations in the Eastern Basin (65-
80 °W) and the Western Basin (15-30 °W) contrasting with the lowest concentrations measured in the central gyre system of
the transect (30-65 °W) (Figure 2). The vertical distribution of CH$_4$ appears to be associated with fluorescence; highest
concentrations of CH$_4$ were found at ~100 m depth, where fluorescence and dissolved oxygen are highest and nutrients levels
lowest, which is in agreement with previous findings (Kudo et al., 2018).

We applied PCA to identify which environmental parameters (nutrients, fluorescence, dissolved oxygen, temperature, salinity,
depth, transmission) could be related to or influence the distribution of CH$_4$ along the transect. 54% of the variability could be



explained by the first component that is primarily controlled by depth (Figure 4). $CH_4$ is mainly associated with oxygenated surface waters, characterised by low concentrations of nutrients and a higher fluorescence. However, the weak contribution of $CH_4$ to the first two components may be due to the heterogeneity of its distribution. To fully understand the relationship between the different parameters controlling the distribution of $CH_4$, we therefore separated the transect into three main regions, i.e. i) the Western Basin, ii) the Central Gyre System and iii) the Eastern Basin. The PCA applied on the regionalized

dataset (Figure 5) revealed that $CH_4$ is clearly associated to the abundance of primary producers in surface waters (<100m) in the western basin, while in the central gyre and eastern basin $CH_4$ concentrations were also influenced either by *in-situ* processes at 400 m (Kudo et al., 2018) or by external inputs from the bottom such as the Mauritanian upwelling that brings $CH_4$-enriched waters to the upper layer (Conrad and Seiler, 1988).





**Figure 4: Principal Component Analysis (PCA) between dissolved CH₄ concentrations (in red) and other physico-chemical parameters (nutrients, depth, fluorescence, temperature, salinity and turbidity). Numbers on the x and y axes indicate the factor loadings of each variable of each principal component (PC). The percentages show the explained variability in the dataset by each PC.**





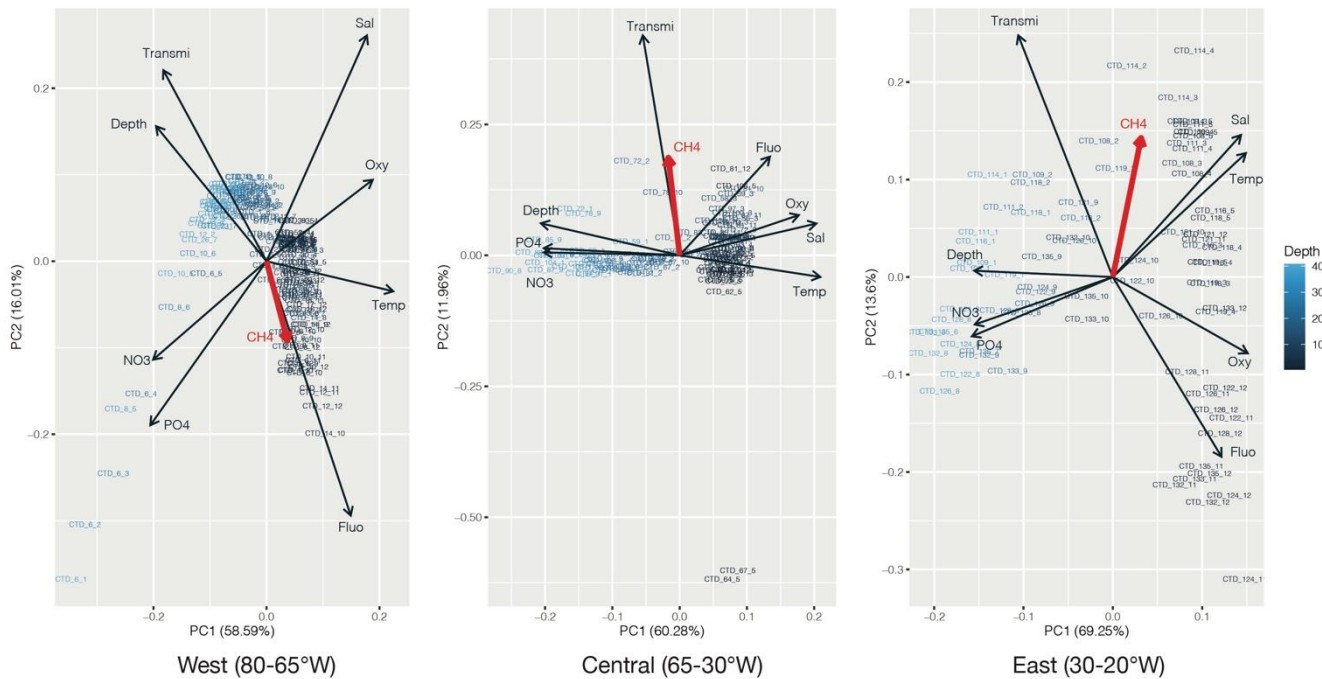

**Figure 5: Principal Component Analysis on the regionalized dataset: west (80-65°W), central (65-30°W) and east (30-20°W). Numbers on the x and y axes indicate the factor loadings of each variable of each principal component (PC). The percentages show the explained variability in the dataset by each PC. CH₄ is highlighted in red.**

Although $CH_4$ appeared to have a uniform vertical distribution (Figure 2), selected profiles from two areas of interest, CTD 50 (69,5 ºW, 24.9 ºN) and CTD 122 (20,8 ºW, 25,45 ºN) of $CH_4$, Chl *a*, phosphate and dissolved oxygen showed that the highest concentration of $CH_4$ in fact correspond to the maxima of Chl *a* and dissolved oxygen and the lowest concentrations of phosphate (Figure 6). The correlation between $CH_4$ and Chl *a* (Kendall rank correlation test, r2=, p<0.05) suggests that the primary producers play a role in the production of $CH_4$.




**Figure 6. Selected data from CTD 50 (69.5 ºW, 24.9 ºN) and CTD 122 (20.8 ºW, 25.45 ºN) of CH₄ (red circles), Chl *a* (green triangles), phosphate (purple squares) and dissolved oxygen (orange).**





At station 50, the relationship between CH$_4$, Chl $a$ and phosphate appears to be linear, e.g. CH$_4$ concentration decreasing as
Chl $a$ decreases and phosphate increases; while at station 122 CH$_4$ concentration showed a nonlinear pattern. It is not clear
why CH$_4$ concentrations are variable but zooplankton grazing could potentially have a substantial impact (Simon et al., 2012).
A possible influence of gas seeps on the CH$_4$ concentration is negligible as gas seeps only influence CH$_4$ concentrations in the
immediate water column 100 to 150 m above the seeps (Leonte et al. 2020). Below 250 m water depth, CH$_4$ concentrations
are decreasing, corresponding to an increase in phosphate and minimal Chl $a$ concentrations, again suggesting the influence
of primary producers (Brown et al., 2014).

## 4.2      Methane production linked to a primary producers

Autotrophic cyanobacteria *Prochlorococcus* and *Synechococcus* represent a major constituent of primary production in the
subtropical North Atlantic, their distribution however differs greatly (Flombaum et al., 2013). Whereas *Synechococcus* solely
occupies surface waters up to depths of ~ 100 m, *Prochlorococcus* occupies the whole water column, with deep-water maxima
just above MLD, therefore responsible for fluorescence maxima at 100 to 130 m. The longitudinal distribution also differs
between both taxa: The distribution of *Synechococcus* is limited to coastal, nutrient richer waters (Figure 7), whereas
*Prochlorococcus* dominates the community throughout the transect (Figure 8).

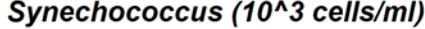

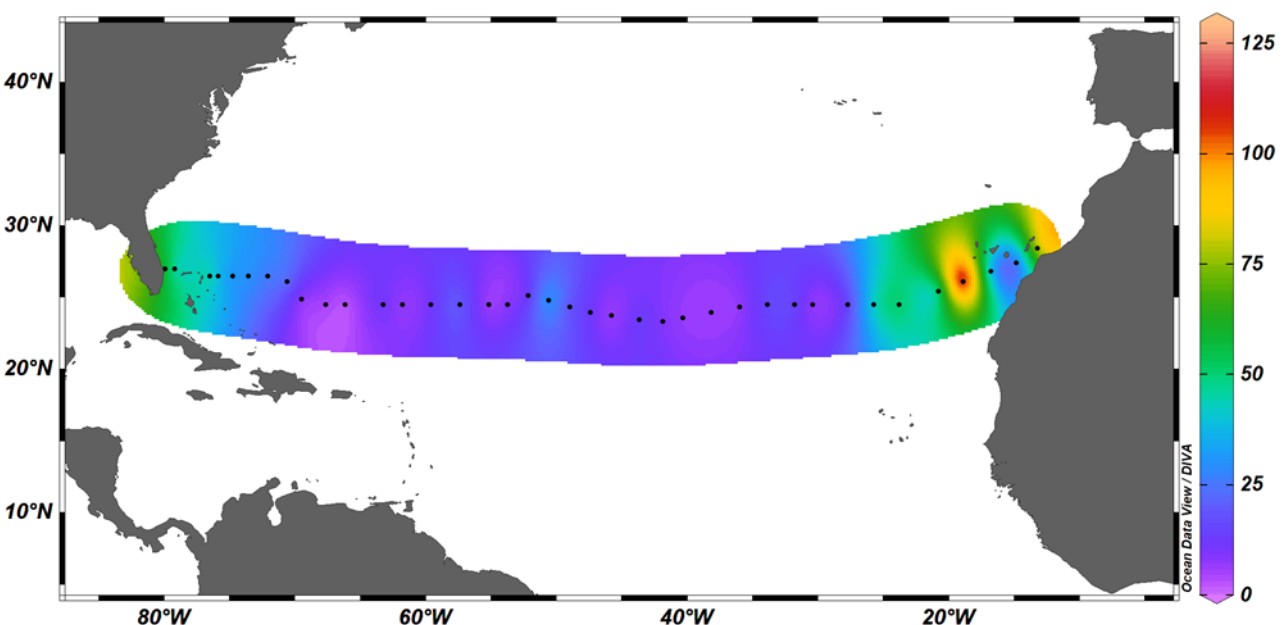




**Figure 7. Depth-integrated abundance of *Synechococcus* (in 10^3 cells/ml) above MLD. Black dots represent the 38 out of the total 135 stations sampled for cyanobacterial abundance.**

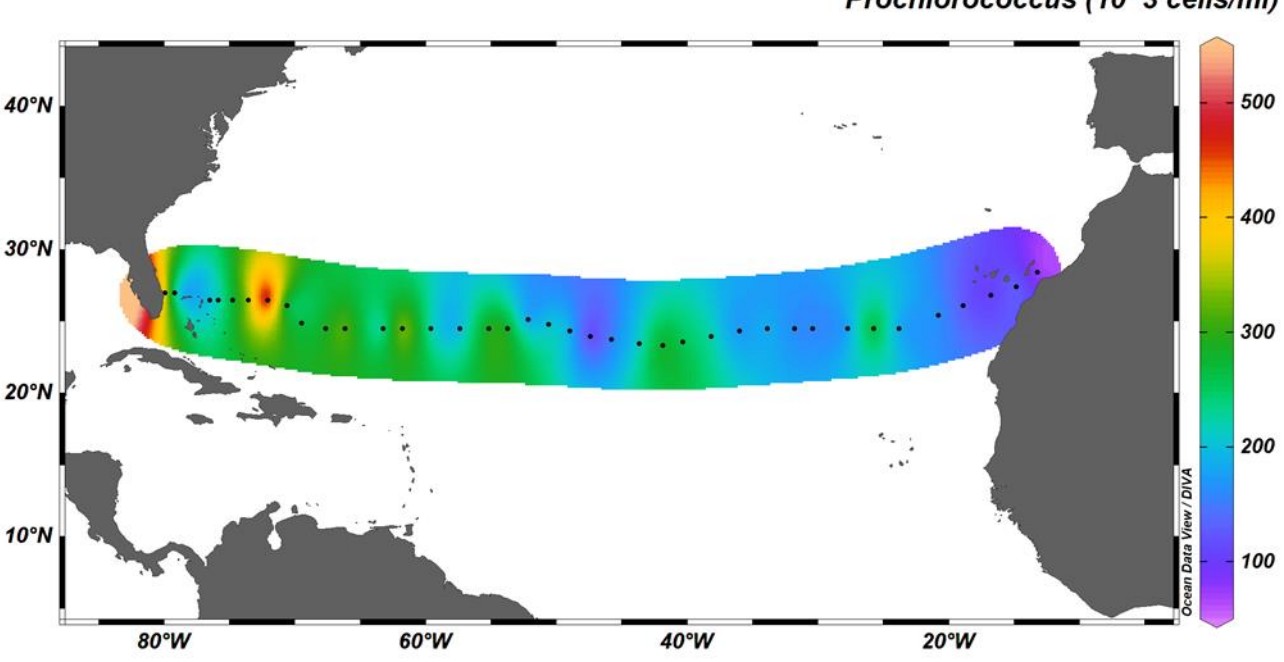


**Figure 8. Depth-integrated abundance of *Prochlorococcus* (in 10^3 cells/ml) above MLD. Black dots represent the 38 out of the total 135 stations sampled for cyanobacterial abundance.**

$CH_4$ distribution across the subtropical north Atlantic suggests the influence of the cyanobacterial community present; highest $CH_4$ concentrations were found 1) at DCM in the central gyre system, where *Prochlorococcus* is the predominant genus and 2) at the gyre boundaries, where higher abundances of both *Prochlorococcus* and *Synechococcus* were found. A PCA applied on the dataset including abundances of cyanobacteria (Figure 9) confirms this trend: CH4 appears to be mainly associated with *Synechococcus* and in a lesser extent with *Prochlorococcus*.







**Figure 9. Principal Component Analysis (PCA) between dissolved CH₄ concentrations (red), *Prochlorococcus* and *Synechococcus* abundances (light blue) and other physico-chemical parameters (nutrients, depth, fluorescence, temperature, salinity and turbidity). Numbers on the x and y axes indicate the factor loadings of each variable of each principal component (PC). The percentages show the explained variability in the dataset by each PC.**





Not only do the bordering Gulf Stream to the West and the Canary Stream to the East provide a resupply of nutrients, and therefore support a greater abundance of cyanobacteria; the Canary Stream presents a pathway for the horizontal transfer of organic matter from the North African coast to the open ocean. Previous studies confirm high OM production in the eastern Subtropical North Atlantic as a result of the high rates of primary production fuelled by the Mauritanian upwelling (Reynolds et al., 2014). The transmission data (Figure 2h) suggests the export of organic matter throughout the eastern basin, which could

contain MPn and thus a potential source of $CH_4$. Elevated $CH_4$ concentrations are mainly limited to the highly productive boundary systems, whilst the central gyre which lacks a sufficient resupply of bioavailable nutrients due to its downwelling nature, results in a decreased abundance of both *Prochlorococcus* and *Synechococcus* and henceforth decreased $CH_4$ concentrations (Figure 2a). In this especially P limited region, alternative nutrient sources such as the degradation of DOM to access organic phosphorus compounds become increasingly important in order for the cyanobacteria to meet their nutrient

needs. The degradation of dissolved organic phosphorus via alkaline phosphatases (AP) such as *phoA* or *phoX* and the overexpression of the phosphate binding protein (pstS) have been believed to be the main adaptations to P-stress (Luo et al., 2009; Cox and Saito, 2013; Sebastian and Ammerman, 2009). Recently however, evidence was brought forth that some strains of *Prochlorococcus* can also oxidise MPn and other higher phosphonate compounds while releasing formate and potentially $CH_4$ as a by-product (Sosa et al., 2019a). Phosphonates are notably abundant and enriched in DOM (Sosa et al., 2019b) and

their degradation releases considerable amounts of $CH_4$. *Prochlorococcus* and *Synechococcus* are the most abundant primary producers in the oligotrophic ocean, and as such produce considerable amounts of semi-labile DOM; both can synthesize phosphoenolpyruvate mutases (*pepM*) and therefore the DOM produced carries an enriched pool of MPn (Repeta et al., 2016; Sosa et al., 2019a). However, the metabolism of MPn is heavily regulated by bioavailable phosphate, thus the metabolic pathway of *pepM* might be heavily down regulated in the subtropical North Atlantic, whereas under replete conditions i.e. in

the North Pacific *Prochlorococcus* can allocate up to 40 % of its internal P-quota towards phosphonate synthesis (Acker et al., 2022). Yet, the trait to produce phosphonates is located on genomic islands and is subject to horizontal gene transfer and can be frequently exchanged among marine microbial communities, hence also proteobacteria such as the SAR11 clade obtain *pepM* and are able to produce phosphonates (Acker et al., 2022). Similarly is the trait of phosphonate consumption subject to horizontal gene transfer and high-light strains of *Prochlorococcus* carry both production Figuand consumption traits, therefore

can also utilise MPn as alternative P source (Acker et al., 2022). Nonetheless, most MPn oxidation and subsequent release of $CH_4$ is due to bacterial degradation of DOM and the breakdown of high energetic carbon-phosphorus bonds via C-P lyases, which are encoded by the *phn* operon, with transport systems including *phnC*, *phnD* and *phonE* and *phnJ* responsible for the cleavage of the C-P bond (Sosa et al., 2020). C-P lyases are abundant among *Pelagibacter spp*. and other alpha and gammaproteobacteria and can be found in ~ 50 % of organisms in the North Atlantic, where DOP concentrations are 4 fold

lower in respect to the North Pacific (Sosa et al., 2020).

Although it remains broadly unclear whether cyanobacteria such as *Prochlorococcus* or *Synechococcus* are producing $CH_4$ mainly via the degradation of MPn or indirectly by producing semi-labile DOM containing MPn cleaved by the bacterial





community, cyanobacteria can greatly influence $CH_4$ production in the marine environment. Furthermore, nitrogen fixing diazotrophs may contribute to $CH_4$ production to a higher degree (Bižić-Ionescu et al., 2018) and are also abundant throughout

the subtropical North Atlantic, with *Trichodesmium spp.* dominating the western basin and *Hemilaulus* associated *Richelia* symbionts more so in the eastern basin (Luo et al., 2009). *Trichodesmium* has higher nutrient requirements and can therefore outcompete cyanobacteria in uptake of inorganic nutrients and degradation of alternative sources; yet, energy intensive diazotrophic nitrogen fixation is controlled by micronutrients such as iron (Macovei et al., 2019) and is primarily occurring in the western basin, an area of high iron input by aeolian plumes originating from the Saharan desert (Ratten et al., 2015;

Reynolds et al., 2014). Lastly, coccolithophores such as the ubiquitous abundant *Emiliania huxleyi* can also produce $CH_4$ from DOM degradation (Lenhart et al., 2016b), and are believed to increase in abundance in the subtropical North Atlantic with increasing $CO_2$ concentrations at the air-sea interface due to further anthropogenic perturbation of the atmospheric $CO_2$ budget (Krumhardt et al., 2015).

Nonetheless, the data presented here suggest that the cyanobacterial community could play a key role in the $CH_4$ flux in surface

waters and the degradation of MPn from semi-labile DOM, contributing to explain the methane paradox and henceforth the sea-air flux of $CH_4$ (Sosa et al., 2019a). Further investigation needs to focus on gathering *in-situ* data and should also include future scenarios, considering future climate and whole ecosystem community responses to consequences of altered climate conditions. The base ecosystem, specifically in P-limited regions is ever changing and highly adapted and horizontal transfer of genomic traits of P acquisition via various genomic pathways might become inherently important.

## 345    5       Conclusions

Our study shows that the subtropical North Atlantic Ocean does indeed act as a source of $CH_4$ to the atmosphere, most likely controlled by cyanobacteria which are the dominant primary producers in the surface waters. Yet, anomalies found at depths below 200 m could also be attributed to the degradation of sinking organic material. The concentrations of dissolved $CH_4$ in this study were considerably higher near shelf regions and in the eastern boundary under the influence of the Mauritanian

upwelling. The accumulation of organic matter and nutrients in these areas provide favourable conditions for both aerobic and anaerobic $CH_4$ production. It is expected that with increasing stratification and subsequent reduction in nutrient supply to the surface oligotrophic North Atlantic Ocean, the prevailing P-limitation will be further exacerbated, whereas coastal and shelf regions with increasing anthropogenic inputs of nutrients, could experience more frequent cyanobacterial blooms which, will in turn enhance $CH_4$ production (Dang and Li, 2018).

## 355    6. Author Contributions

Anna Kolomijeca – carried out dissolved gas samples collection, measurements and data analysis of the dissolved gases included methane and drafted the final manuscript



Lukas Marx – carried out phytoplanktonic sample collection, measurements and data analysis

Sarah Reynolds – supervised biological part of the experiments

Thierry Cariou - carried out nutrient distribution study

Edward Mawji - carried out nutrient distribution study

Cedric Boulart – helped to plan and prepare for the cruise, helped with results evaluation and provided overall project supervision

All authors contributed equally to the manuscript writing.

**7. Conflict of Interest**

The authors declare that the research was conducted in the absence of any commercial or financial relationships that could be construed as a potential conflict of interest.

**8. Funding**

Shiptime was funded by NERC as part of the Climate Linked Atlantic Sector Science (CLASS) research programme and as a
UK contribution to the Global Ocean Ship-based Hydrographic Investigations Program (GO-SHIP).

A Kolomijeca's participation to the cruise as well as dissolved gas analyses were funded by the Region Bretagne SAD 'FULMAR' project, the Fondation Air Liquide SMIS-4M project and the CNRS (LEFE Memestra).

L. Marx's and Sarah Reynolds' participation was funded by the University of Portsmouth PhD bursary scheme and a UK NERC National Capability programme CLASS (Climate Linked Atlantic Sectore Science) ECR fellowship.


**9. Acknowledgments**

The Authors authors gratefully acknowledge the PI of the JC191 cruise Alejandra Sanchez-Franks (National Oceanography Centre, UK) for the great scientific support and providing of the cruise data. This work would not have been possible without the support and help of the Captain and crew of RRS James Cook. The authors would also like to acknowledge Eric Mace
(Station Biologique de Roscoff) – the CNRS technician for his technical support in the preparation of the cruise.

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
