# Peer review of "An update on dissolved methane distribution in the North subtropical Atlantic Ocean"

_EGUsphere, 2022_

## Referee Comment (RC1)

**Peer review: An update on dissolved methane distribution in the North subtropical Atlantic Ocean**

**General comments**

This manuscript by Kolomijeca et al. addresses the environmentally important ocean methane paradox (OMP) and both widens and confirms previous studies in this field. Most importantly, their work seems to be rooted on a solid dataset, comprising many parameters, thereby allowing interesting insights into this phenomenon. It also seems to be clear that the two investigated cyanobacterial organisms significantly contribute to methane formation in the marine environment. Still, it does not become fully clear how great the contribution of these organisms to the overall OMP, in comparison to other marine species, really is – this point should be more openly addressed (also if a clear answer cannot be provided). Furthermore, spelling/grammar of the manuscript require minor improvements. Overall, this manuscript provides important field data that (i) confirm and describe the OMP more closely and (ii) thereby help to provide a solid base for future, more mechanistic studies.

**Specific comments**

L78: a bit hard to understand. Do you mean methyl radicals? If not, how actually are methyl groups converted to methane? This should be a bit more precise.

L138ff: For flow cytometry, the gating strategy including forward/side scattering should generally be provided as a figure or in the appendix (in many journals, this is also a strict requirement). Especially in this specific case, this would make a lot of sense, as the distinction between two cyanobacterial species is facilitated by this method.

Figure 2: Can you abbreviate dissolved oxygen with "D"? Technically of course, but another abbreviation like "$O_2$" would be a bit more intuitive? And in order to put all these data into context, could the x-axis contain a bit more information about the respective area (e.g., could you highlight the mid-ocean-ridge there?). Additionally, the depicted depth here always goes down until 400m – you mentioned before that some measurement stations were less deep than 400m, how does this relate? Otherwise interesting dataset.

L225: Although shortly mentioned in some sections before, the parameter "fluorescence" is a bit unexpected here – I think the readability would significantly improve if you again shortly describe what the fluorescence actually indicates here. Additionally, Fig. 2 could mention/depict the fluorescence parameter?

L232: To me, it is unclear what the mentioned "first two components" are – depth, nutrients, fluorescence…? Also, does "weak contribution" mean that these (unclear) two components

strongly correlate with methane levels, although methane levels do not really correlate with these two components? I do not fully understand this sentence / section, I think it should be (partially) rewritten.

L236/37: Could you give an example for *in-situ* processes?

L250: "uniform vertical distribution" -> maybe a slight overinterpretation. You should also explain why you selected these two areas of interest and why it made scientifically sense to choose these. Otherwise, the subsequent results/interpretations can be questioned.

L290ff: Fig. 9 -> the mentioned physico-chemical parameters in L291 in brackets do not relate to the parameters presented in the corresponding Figure (e.g. turbidity, salinity mentioned in the text, but not shown in the figure), please clarify.

L328ff: "nitrogen-fixing diazotrophs may contribute to $CH_4$ production to a higher (!) degree" -> so how do the methane amounts formed by *Prochlorococcus* and *Synecoccus* relate to methane amounts formed by other marine bacteria in the MLD? Do these nitrogen-fixing strains or coccolithophores overall form more methane? While I agree that a precise quantitative comparison is impossible, I would be at least interested how the orders of magnitude of formed methane relate to each other between these species. Along these lines, I also believe that the authors do not provide the reader with a clear estimation, to which extent the measured $CH_4$ levels are actually caused by *Prochlorococcus* and *Synecoccus* and to which extent by other organisms – I am not sure whether this information can be extracted from the dataset, but, at least, it would be helpful to clearly address this open question. Right now, the main message of this manuscript simply seems to be that these cyanobacteria belong to the important producers of methane in marine environments.

L328ff: Apart from the MPn/C-P-Lyase pathway, it should be clarified whether the other organisms mentioned (e.g. *E. huxleyi*) might also be able to form methane via alternative mechanisms.

**Technical corrections**

Title: Rather "subtropical North Atlantic Ocean" instead of "North subtropical Atlantic Ocean"?

L12/13: Rephrase first sentence, second comma after "however" necessary as also done in L30 (2 sentences might be better though)

L19: phosphorus depleted -> change to phosphorus-depleted

L27: Delete comma after "Earth warming"

L28: change "..increased by 20% [..] –and is expected.." to "increased by 20% and ARE expected.."

L28/29: slightly rephrase and delete comma, e.g. "..expected to further rise by approx…"

L31: CH4 -> $CH_4$ (subscript)

L33-35: This sentence should be slightly rephrased, generally avoid ";"

L35: Add comma after "pressure and temperature"

L37-39: This sentence might be split into two sentences

L40: Bracket is missing, also "(up to 75 %)" should not be in brackets

L43: Either "THE typical $CH_4$.." or "…depth distributions indicate"

L44: Replace ";" with "."

L47: Remove comma before "where"

L61: Remove ";" -> change to "*Synechoccus*. In fact, the oligotrophic…"

L75: A citation concerning environmental/oxidative stress would maybe help here

L77: "Oceans" -> "Ocean"

L80 and 81: CH4 -> $CH_4$ (subscript), "phosphate-replete"

L82: Check spelling of DMSP in brackets

L89: "monitoring and investigating" -> remove both "the" before and "of" afterwards

L109: into -> in?

L111: Poisoned? Another word might be better?

L112: on shore -> onshore

L124: Sub/superscript

L128: Remove one bracket

L155: add comma after "coefficient"

L184: add "levels" after "Chlorophyll a", remove comma

L205: Remove comma after "note"

L206: Location-specific

L208: "Sea-to-air"

L209. However,

L262: Comma after "negligible"

L273: nutrient-richer

L284: subtropical North Atlantic

L287: CH4 -> $CH_4$ (subscript)

L299: suggest

L301: comma after "gyre"

L303: P-limited

L303ff: The degradation of DOM is not a source itself but a process, please rephrase this sentence

L311: Remove ";", start new sentence

L314: downregulated

L319: Figuand?

L324: 4-fold

---

## Author Response (AR1)

Reviewer 1

This manuscript by Kolomijeca et al. addresses the environmentally important ocean methane paradox (OMP) and both widens and confirms previous studies in this field. Most importantly, their work seems to be rooted on a solid dataset, comprising many parameters, thereby allowing interesting insights into this phenomenon. It also seems to be clear that the two investigated cyanobacterial organisms significantly contribute to methane formation in the marine environment. Still, it does not become fully clear how great the contribution of these organisms to the overall OMP, in comparison to other marine species, really is – this point should be more openly addressed (also if a clear answer cannot be provided). Furthermore, spelling/grammar of the manuscript require minor improvements. Overall, this manuscript provides important field data that (i) confirm and describe the OMP more closely and (ii) thereby help to provide a solid base for future, more mechanistic studies.

Response to Reviewer 1:

We thank the reviewer for this feedback and appreciate the concerns regarding the individual contribution of both *Prochlorococcus* and *Synechococcus* to OMP. The authors note, that the scope of this work was not to measure and quantify the specific methane production by these two prominent primary producers, but to correlate their occurrence and distribution to areas of interest in regards to oceanic methane production in order to highlight the involvement of the cyanobacterial community towards the OMP. Furthermore, we provide insight into the metabolic mechanisms involved in methane production and relate these to concurring nutrient regimes present in the area of study. However, we agree, that there was a need to further discuss the individual contribution of different primary producers to oceanic methane production and we now include a paragraph in the discussion presenting individual rates of methane production from other studies for different primary producers, including both mentioned in the present manuscript and other abundant groups. The authors would like to mention, that to measure individual methane production rates in situ is rather difficult and requires additional experimental design and setup, i.e. mesocosms and/or sensor-controlled incubation settings and the distinction of individual rates per organism would be further complicated when using natural communities. As stated by the reviewer, our work provides a solid base for future research of the OMP, which should focus on such further developed experimental studies to estimate individual rates of different marine organisms. Here, we present observational high-resolution data to highlight the need for further research regarding OMP. Lastly, we improved the language of our manuscript and hope that we could satisfy the reviewer's comments with the revised version. We would like to thank the reviewer again and hope that the revised manuscript will be considered for acceptance.

On behalf of all authors,

Anna Kolomijeca

Reviewer 2

I support the publication of this paper.  The observations and analysis discussed in this paper were clearly described and the results support the conclusions drawn.  The overall objective of the manuscript "An update on dissolved methane distribution in the North subtropical Atlantic Ocean" by Kolomijeca et al. is good and is a useful contribution to the understanding of the release of methane from the oceans or the ocean methane paradox (OMP) and the factors that control the concentration distribution and the flux of $CH_4$.  The results discussed in the paper support the hypothesis that cyanobacteria play a role in regulating $CH_4$ concentrations in upper waters with higher oxygen concentrations.  The types of measurements made and the analysis completed are very good.  Additional work needs to be done to show that the mechanisms suggested are really occurring in the natural environment.  This paper supports this direction of future research.  There were only a few minor typos.

Response to Reviewer 2:

The authors would like to thank the reviewer for this feedback and are pleased that it was received well. We appreciate, that the reviewer shares our view that cyanobacteria might play a significant role towards the OMP. We do agree, that additional work is needed, both in investigating the occurrence of the mechanisms of methane production, described in this manuscript, in a natural environment and also to estimate specific rates of methane production in natural communities. However, this was not the scope of this work, but to provide a fundamental understanding of the role cyanobacteria play in the OMP. In our revised manuscript, we do now include a paragraph discussing individual rates of methane production of different marine primary producers, including both mentioned in our paper. We note, that there clearly is need for further research and would like to provide a basis with this manuscript present and enable research to proceed in this direction. Further, we did appreciate the reviewers comment regarding the language and worked over the manuscript. The authors would like to thank the reviewer again for his feedback and hope that the revised manuscript will be satisfactory and considered for publication.

On behalf of all authors,

Anna Kolomijeca